# Impact of Δ^9^-Tetrahydrocannabinol on Rheumatoid Arthritis Synovial Fibroblasts Alone and in Co-Culture with Peripheral Blood Mononuclear Cells

**DOI:** 10.3390/biomedicines10051118

**Published:** 2022-05-11

**Authors:** Torsten Lowin, Christina Kok, Sophie Smutny, Georg Pongratz

**Affiliations:** Poliklinik, Funktionsbereich & Hiller Forschungszentrum für Rheumatologie, University Hospital Duesseldorf, 40225 Duesseldorf, Germany; christina.Kok@uni-duesseldorf.de (C.K.); sophie.Smutny@hhu.de (S.S.); georg.pongratz@hhu.de (G.P.)

**Keywords:** synovial fibroblast, rheumatoid arthritis, cannabis, tetrahydrocannabinol, cytokines, calcium, PBMC, cannabinoid receptors, TRPA1

## Abstract

δ9-Tetrahydrocannabinol (THC) has demonstrated anti-inflammatory effects in animal models of arthritis, but its mechanism of action and cellular targets are still unclear. The purpose of this study is to elucidate the effects of THC (0.1–25 µM) on synovial fibroblasts from patients with rheumatoid arthritis (RASF) and peripheral blood mononuclear cells (PBMC) from healthy donors in respect to proliferation, calcium mobilization, drug uptake, cytokine and immunoglobulin production. Intracellular calcium and drug uptake were determined by fluorescent dyes Cal-520 and PoPo3, respectively. Cytokine and immunoglobulin production were evaluated by ELISA. Cannabinoid receptors 1 and 2 (CB_1_ and CB_2_) were detected by flow cytometry. RASF express CB_1_ and CB_2_ and the latter was increased by tumor necrosis factor (TNF). In RASF, THC (≥5 µM) increased intracellular calcium levels/PoPo3 uptake in a TRPA1-dependent manner and reduced interleukin-8 (IL-8) and matrix metalloprotease 3 (MMP-3) production at high concentrations (25 µM). Proliferation was slightly enhanced at intermediate THC concentrations (1–10 µM) but was completely abrogated at 25 µM. In PBMC alone, THC decreased interleukin-10 (IL-10) production and increased immunoglobulin G (IgG). In PBMC/RASF co-culture, THC decreased TNF production when cells were stimulated with interferon-γ (IFN-γ) or CpG. THC provides pro- and anti-inflammatory effects in RASF and PBMC. This is dependent on the activating stimulus and concentration of THC. Therefore, THC might be used to treat inflammation in RA but it might need titrating to determine the effective concentration.

## 1. Introduction

*Cannabis sativa* contains over 400 compounds, with tetrahydrocannabinol (THC) and cannabidiol (CBD) being the most thoroughly investigated [1]. THC binds to classical cannabinoid receptors (CB_1_ and CB_2_) with high affinity but also targets other receptors, enzymes and transporters at higher concentrations [2,3,4]. While CB_1_ is the most abundant G-protein-coupled receptor in the brain [5], CB_2_ is mainly located peripherally [6,7]. Anti-inflammatory effects of cannabinoids occur directly via CB_2_ activation [8], but also by modulation of sympathetic nervous system activity [9]. THC, an agonist at CB_1_ and CB_2_, resembles the effect of endogenous cannabinoids (endocannabinoids), which are produced by a wide variety of cells [10]. Of note, endocannabinoids control the tone of nervous system activity, including the sympathetic nervous system [11]. THC mediates its effects via several mechanisms: (1) directly on cells that carry target receptors for THC (e.g., CB_1_ and CB_2_) [12], (2) indirectly by modulation of neurotransmitter release via CB_1_ on nerve terminals (e.g., acetylcholine and norepinephrine) in the periphery [13], and (3) indirectly by binding to fatty acid binding proteins, which releases endocannabinoids from these binding sites and increases their endogenous concentration [4]. THC has analgesic properties, but also demonstrates antiedema and anti-inflammatory effects [14,15,16,17,18] in animal models of arthritis. However, it is still unclear what receptors and pathways are engaged by THC to elicit these effects.

Rheumatoid arthritis (RA) is an autoimmune inflammatory disorder that is characterized by joint destruction, elevated cytokine burden and several comorbidities such as depression, cachexia, insulin resistance and fatigue [19,20,21,22]. Cannabinoids might reduce pain and inflammation in RA [15,16] but they might also have beneficial effects on RA comorbidities, since these are often mediated by alterations in sympathetic nervous system activity [23,24,25,26]. One major mediator of joint pathology in RA is synovial fibroblasts (SF), which not only produce cytokines and matrix metalloproteinases, but also actively engage in cartilage invasion [27]. Rheumatoid arthritis synovial fibroblasts (RASF) express cannabinoid receptors, and activation of CB_1_ or CB_2_ initiates MAP kinase signaling [28]. While CB_1_ mediates adhesion of RASF to extracellular matrix [29], CB_2_ activation was described as anti-inflammatory in several studies [30,31] but, in contrast, it was also identified as proinflammatory by initiating transforming growth factor beta-activated kinase 1 kinase signaling [32]. In addition, inhibitory effects of synthetic cannabinoids on cytokine production were cannabinoid-receptor-independent [31,33]. In peripheral blood mononuclear cells (PBMC), similar results regarding the effects of cannabinoids have been documented. Springs et al. showed a reduction in splenocyte cytokine production by THC, which was independent of cannabinoid receptors [34]. CB_2_, in general, affects many aspects of the immune response but, in most models of experimental arthritis in rodents, CB_2_ reduces inflammation and arthritis severity by inhibiting immune cell migration to sites of inflammation [35,36,37,38,39].

In this study, we investigate the effects of THC (0.1–25 µM) on RASF in respect to intracellular calcium levels, drug uptake, proliferation and cytokine production. In addition, we perform RASF/PBMC co-cultures and PBMC monocultures under the influence of THC (1 and 10 µM) and determine cytokine and immunoglobulin production. This study might help to pinpoint the effects of cannabis and THC on arthritic disease and establish a rationale for why medical cannabis might be an appropriate adjunct therapy in RA.

## 2. Materials and Methods

### 2.1. Patients

A total of 14 patients with long-standing RA fulfilling the American College of Rheumatology revised criteria for RA [40] were included in this study. The RA group comprised of 14 females with a mean age of 68 years ± 10 years. C-reactive protein was 7.1 mg/dL± 9.2 mg/dL 3 out of 14 glucocorticoids, 4 out of 14 methotrexate, 1 out of 14 sulfasalazine, 3 out of 14 biologicals and 1 out of 14 JAK inhibitor. All patients underwent elective knee joint replacement surgery, and they were informed about the purpose of the study and gave written consent. The study was approved by the Ethics Committees of the University of Düsseldorf (approval numbers 2018-87-KFogU and 2018-296-KFogU). We confirm that all experiments were performed in accordance with relevant guidelines and regulations.

### 2.2. Compounds

THC (Dronabinol) was obtained from THC Pharm, Frankfurt, Germany. A967079, COR170, HC030031, ruthenium red (RR) and rimonabant were obtained from Tocris/Biotechne, Wiesbaden, Germany. 1,1′-Diethyl-2,2′-cyanine iodide (Decynium-22; D22) was obtained from Sigma Aldrich, Taufkirchen, Germany. THC is a partial agonist at CB_1_ and CB_2_ receptors, but also at several TRP channels. It is the major psychoactive constituent of the plant *cannabis sativa*. A967079 and HC030031 are lipophilic antagonists at the TRPA1 ion channel, whereas RR unselectively blocks TRP channels at the plasma membrane. Rimonabant and COR170 are inverse agonists at CB_1_ and CB_2_, respectively. D22 is an inhibitor of organic cation transporters and monoamine transport.

### 2.3. Synovial Fibroblast and Tissue Preparation

Samples from RA synovial tissue were collected immediately after opening the knee joint capsule, and tissue was prepared for cell isolation thereafter [41]. Synovial tissue was cut into small fragments and treated with liberase (Roche Diagnostics, Mannheim, Germany) at 37 °C overnight. The cell suspension was filtered (70 µM) and centrifuged at 300 g for 10 min. After that, the pellet was treated with erythrocyte lysis buffer (20.7 g NH_4_Cl, 1.97 g NH_4_HCO_3_, 0.09 g EDTA ad 1 L H_2_O) for 5 min, recentrifuged for 10 min, and then resuspended in RPMI-1640 (Sigma Aldrich, Taufkirchen, Germany) with 10% FCS. After overnight incubation, RPMI medium was replaced with fresh medium to wash off dead cells and debris.

### 2.4. Intracellular Calcium and PoPo3 Uptake

In black 96-well plates, RASF were incubated with 4 µM of calcium dye Cal-520 (ab171868, Abcam, Cambridge, UK) in PBS with 0.02% Pluoronic F127 (Thermo Fisher scientific, Waltham, MA, USA, # P6866) for 60 min at 37 °C, followed by 30 min at room temperature. After washing, HBSS or PBS containing 1 µM PoPo3 iodide (Thermo Fisher scientific, # P3584) and respective antagonists/ligands/inhibitors were added for 30 min at room temperature. After that, THC was added and the intracellular Ca^2+^ concentration as well as PoPo3 uptake were evaluated with a TECAN multimode reader over 90 min.

### 2.5. Flow Cytometry

RASF were primed with TNF (10 ng/mL) (PeproTech, Hamburg, Germany) or left untreated for 72 h in RPMI medium with 2% FCS. Then, cells were analyzed for surface and intracellular expression of cannabinoid receptors. The following antibodies were used: CB_1_ (FAB3834R, 0.2 mg/mL, 1:10, R&D Systems/Biotechne, Wiesbaden, Germany), CB_2_ (FAB36551G, 0.2 mg/mL, 1:40, R&D Systems/Biotechne), Isotype MsIgG2a-Alexa 488 (IC003G, 5 µL/test, R&D Systems/Biotechne), and Isotype MsIgG2a-Alexa 647 (IC003R, 5 µL/test, R&D Systems/Biotechne5 µL/test); RASF were detached from culture dishes with citrate buffer (135 mM KCl, 15 mM Na_3_C_6_H_5_O_7_) and centrifuged at 300× *g*. Cells were resuspended in PBS with 10% FCS and incubated with antibodies for 30 min in the dark at room temperature. For intracellular staining, the inside stain kit was used (#130-090-477, Miltenyi biotec, Bergisch Gladbach, Germany) according to the manufacturer’s instructions.

### 2.6. Isolation of PBMC from Peripheral Blood

PBMC were isolated using the Greiner LeucoSep Tubes (#227290, Greiner bio-one, Kremsmünster, Austria) according to manufacturer’s instructions.

### 2.7. RASF Co-Culture with PBMC

Co-culture experiments were performed in 96-well plates (Cellstar, Greiner bio-one, Kremsmünster, Austria). In brief, 5.000 RASF were seeded in 200 µL RPMI-1640 with 10% FCS (Sigma-Aldrich) and grown for 72 h. Then, growth medium was replaced by fresh RPMI with 10% FCS, and 250.000 isolated human PBMCs were added. Cells were stimulated with cytokines/THC as indicated for 7d in RPMI medium with 10% FCS. After that, supernatants were collected and cytokine and immunoglobulin production were assessed by ELISA.

### 2.8. ELISA and Stimulation of SF

ELISAs for IL-6 (#555220), IL-10 (#555157) and TNF (#555212) were obtained from BD, Franklin Lakes, NJ, USA and were conducted according to the manufacturer’s protocol. Immunoglobulin M (IgM) and G (IgG) were detected by an in-house ELISA. A total of 5.000 RASF were seeded in 200 µL RPMI-1640 with 10% FCS and grown for 72 h. Then, growth medium was replaced by fresh RPMI (2% FCS) and SF were primed with TNF (10 ng/mL) for 3 days to induce TRPA1 protein. After that, culture medium was replaced with RPMI (2% FCS) and THC was added for an additional 24 h. After that, supernatants were collected and analyzed.

### 2.9. RASF Cell Viability

Cell viability was assessed by the cell titer blue viability assay (Promega, Madison, WI, USA, # G8080) according to manufacturer’s instructions.

### 2.10. Statistical Analysis

Statistical analysis was performed with SPSS 25 (IBM, Armonk, NY, USA). The statistic tests used were chosen according to previous reports and are given in the figure legends [42,43,44]. Normal distribution was determined using the Shapiro–Wilk test; equal variance was determined by Levene’s test. In the case of equal variance, the Bonferroni post hoc test was used, otherwise the Dunnet’s post hoc test was employed. When data are presented as box plots, the boxes represent the 25th to 75th percentiles, the lines within the boxes represent the median, and the lines outside the boxes represent the 10th and 90th percentiles. When data are presented as line plots, the line represents the mean. When data are presented as bar charts, the top of the bar represents the mean and error bars depict the standard error of the mean (sem). The level of significance was *p* < 0.05.

## 3. Results

### 3.1. RASF Express CB_1_ and CB_2_

In our experiments, RASF were treated with TNF (10 ng/mL) for 72 h before we conducted our experiments and, although the expression of CB_1_ and CB_2_ in SF was already documented [28], their regulation by TNF was only investigated by our group but under different experimental conditions [45]. We found little CB_1_ expression at the cell surface but high intracellular levels that were not significantly regulated by TNF (Figure 1). CB_2_ was exclusively found at the plasma membrane and it was upregulated by TNF (*p* = 0.05) (Figure 1).

### 3.2. THC Increases Intracellular Calcium in RASF Primed with TNF

Target receptors for THC include TRP ion channels [3] and CB_1_ and activation [46] of either is coupled to elevations or reductions in intracellular calcium levels, respectively. Without TNF priming, THC (0.1–25 µM) did not modulate intracellular calcium levels (Figure 2A). However, when RASF were treated with TNF 72 h prior to THC addition, we detected a significant increase (up to ~200%) in intracellular calcium levels in response to THC (*p* < 0.001; 5–25 µM) (Figure 2B). This increase was not inhibited by the CB_1_ antagonist/inverse agonist rimonabant (Figure 2C) and slightly modulated by the CB_2_ antagonist COR170 (Figure 2D). TRPA1 is strongly upregulated by TNF [47] and, since it is also a receptor for THC, we inhibited this channel with ruthenium red (RR) (Figure 2E). RR not only increased basal intracellular calcium levels (*p* < 0.001), but also reduced THC-induced intracellular calcium levels (5 µM and 10 µM THC; *p* = 0.032 and *p* < 0.001, respectively). Previous results from our group suggest that TRPA1 is located intracellularly [45,47] and, since RR cannot actively cross the plasma membrane [48], we also employed specific lipophilic TRPA1 antagonists (Figure 2F,G). We found that both HC030031 (Figure 2F) and A967079 (Figure 2G) inhibited the stimulatory effects of THC (*p* < 0.001 for 5 µM and 10 µM THC (Figure 2F); *p* < 0.001 for 5 µM–25 µM THC (Figure 2G)) on intracellular calcium levels over a wide range of THC concentrations. The latter was more potent, since, in contrast to HC030031, it also reduced calcium elevations in response to the highest concentration of THC (25 µM, *p* < 0.001). We also conducted these experiments with PBS (Figure 2H–M) instead of HBSS, establishing a calcium-free extracellular environment. Under these conditions, alterations in intracellular calcium levels can only be elicited by emptying intracellular stores. Similar results compared to the HBSS groups were obtained but, under these conditions, the TRPA1 antagonist A967079 (Figure 2M) completely abrogated all effects of THC on intracellular calcium.

### 3.3. THC Enhances PoPo3 Uptake in RASF Primed with TNF

In a previous study, we established PoPo3 as a surrogate marker for drug uptake [47], which was coupled to intracellular calcium levels and, therefore, we also assessed the ability of THC (0.1–25 µM) to modulate PoPo3 uptake. Without TNF priming, PoPo3 uptake was only slightly increased by THC (Figure 3A), whereas, after TNF stimulation, THC robustly increased PoPo3 uptake (Figure 3B; *p* < 0.001 for [c] 1 µM−25 µM of THC). Rimonabant modulated PoPo3 uptake only at low THC concentrations (0.1–1 µM), but the magnitude of uptake at these concentrations was rather small (Figure 3C). The CB_2_ inverse agonist COR170 also reduced PoPo3 uptake by THC (Figure 3D; *p* < 0.001; 1 µM, 10 µM, 25 µM THC). RR modulated PoPo3 levels to all but the highest concentration of THC (Figure 3E), but increased rather than decreased its uptake. Like intracellular calcium, PoPo3 uptake was almost completely inhibited by the TRPA1 antagonists HC030031 (Figure 3F; *p* < 0.001 for all [c] of THC except 0.1 µM) and A967079 (Figure 3G, *p* < 0.001 for all [c] of THC except 0.1 µM). Decynium-22 (D22), an inhibitor of organic cation transporters [47], inhibited PoPo3 uptake alone (Figure 3H, blue line, *p* = 0.018) and together with THC (Figure 3H, *p* < 0.001 for all [c] of THC). Lastly, we investigated whether THC itself can block subsequent effects of added THC in higher concentrations and we found that it indeed inhibited further PoPo3 uptake by higher concentrations of THC (Figure 3I; *p* < 0.001, 10 µM and 25 µM). We also assessed the ability of THC to induce PoPo3 uptake without extracellular calcium in PBS (Figure 3J–Q). We confirmed our findings from the HBSS groups, but the CB_2_ antagonist COR170 showed a higher efficacy in calcium-free conditions (Figure 3L; *p* < 0.001, for all [c] of THC). RR, HC030031, A967079 and D22 also inhibited Popo3 uptake almost completely (Figure 3M–O; *p* < 0.001 for all [c] of THC, except 0 µM and 0.1 µM in the A967079 and D22 group). THC itself also reduced PoPo3 uptake, but the effect was attenuated compared to the conditions with extracellular calcium (Figure 3Q).

### 3.4. THC Reduces Cytokine Production Only at High Concentrations

Besides intracellular calcium and PoPo3 uptake, we assessed whether THC (0.1–25 µM) also modulates cytokine production by RASF. We identified TRPA1 as an important target receptor for THC and, therefore, we induced its expression by stimulating RASF for 72 h with TNF before adding THC. THC did not modulate IL-6 or IL-8 production significantly but it blunted MMP-3 levels either alone or in combination with A967079 or rimonabant at 25 µM (Figure 4C, *p* < 0.001). IL-8 production was only reduced by THC (25 µM) when combined with A967079 (*p* < 0.001) or rimonabant (*p* = 0.005). Cell viability was slightly enhanced by THC at 5 µM (*p* = 0.022) but extensive cell death occurred at 25 µM (*p* < 0.001) (Figure 4D). In addition, cell viability was slightly increased when THC (1 µM, 5 µM and 10 µM) was combined with rimonabant or A967079, but the magnitude was small.

### 3.5. THC Has Negligible Effects on PBMC Cytokine, IgM and IgG Production

In synovial tissue, endocannabinoids are abundantly produced not only by RASF, but immune cells are also capable of producing anandamide and 2-AG [28,29,49]. Since lymphocytes and macrophages are also present in RA synovial tissue where these cells closely interact with RASF [27], we investigated the impact of THC (1 and 10 µM) on peripheral blood mononuclear cells (PBMC) alone or in co-culture with RASF (Figure 5). In co-culture with RASF, THC did not modulate IL-6 and IL-10 production but decreased TNF production when PBMC/RASF were stimulated with CpG or IFN-γ (1 µM THC, *p* = 0.027 and *p* = 0.010, respectively) (Figure 5C). Immunoglobulin G production induced by CpG was further enhanced by 1 µM and 10 µM THC, but it did not reach significance (*p* = 0.077 and *p* = 0.085, respectively) (Figure 5E). Without RASF, 10 µM THC reduced IL-10 levels in response to IFN-γ (*p* = 0.011) and CpG (*p* = 0.03) in PBMC (Figure 5G). Immunoglobulin G production was fostered by 1 µM THC without any additional stimulus (*p* = 0.026) (Figure 5J).

## 4. Discussion

In this study, we show for the first time the effects of THC treatment on the function of RASF and PBMC. Firstly, we determined the expression of the main target receptors for THC, but also for endocannabinoids produced in the joint [28], CB_1_ and CB_2,_ in and on RASF, and found CB_1_ in intracellular compartments, whereas CB_2_ was located at the plasma membrane. The latter was also upregulated by TNF. However, we demonstrated that THC elevates intracellular calcium and PoPo3 uptake by a TRPA1, rather than a CB receptor-dependent mechanism. RASF IL-6, IL-8 and MMP-3 production was reduced by THC only at the highest concentration investigated, and this effect was not antagonized by either TRPA1 or CB_1_ antagonists, suggesting a receptor-independent effect. In addition, we showed that THC had only a minor influence on PBMC (alone or in co-culture with RASF) cytokine and immunoglobulin production.

We detected CB_1_ protein intracellularly in RASF, which is line with our previous results [45]. In that study, we utilized a different antibody but it also detected CB_1_ protein at the nuclear membrane, suggesting that signals initiated by this receptor are spatially confined, as already confirmed by other intracellular G-protein-coupled receptors (GPCRs) [50]. Another study showed that membrane GPCRs might “communicate” with its fraction of intracellular receptors by β-arrestin eliciting a combined response [51]. This might also be the case in RASF, since we found little but detectable cell surface expression of CB_1_. CB_2_ was exclusively located at the plasma membrane and its levels were regulated by TNF. In a previous study, we already demonstrated CB_2_ upregulation in response to TNF, but these experiments were conducted under hypoxic conditions and CB_2_ was detected by cell-based ELISA rather than flow cytometry [45].

In a next step, we investigated the effects of THC on intracellular calcium since it has been shown that CB_1_, CB_2_ and TRPA1 are able to modulate calcium levels [52,53,54]. We found that THC via TRPA1 activation elevates calcium levels and we already demonstrated that only TNF pre-stimulated RASF responded to TRPA1 agonism [54], which was confirmed in this study. In contrast to cannabidiol (CBD), THC is specific for TRPA1 and does not engage in significant off-target effects. Although both THC and CBD bind to TRPA1, CBD increased intracellular calcium levels mainly by disrupting mitochondrial calcium homeostasis with little TRPA1 contribution [47].

In previous studies, we revealed that elevations in intracellular calcium are coupled to the uptake of the cationic dye PoPo3 [47,54]. Therefore, we also investigated this uptake in response to THC and we found that intracellular PoPo3 levels increased. PoPo3 was identified as a surrogate marker for drug uptake [47], since its uptake is likely controlled by organic cation transporters that are also responsible for the uptake of several therapeutic compounds and drugs [47]. This finding might be relevant in RA therapy, since a combination of THC with an antiarthritic compound might have synergistic effects and, due to increased cellular uptake, lower drug concentrations might be necessary to elicit similar effects. In fact, this has already been demonstrated with CBD, and, in their study, the authors showed a higher efficacy of the chemotherapeutic drug doxorubicin when combined with CBD [55].

TRPA1 was identified as target receptor of THC and this channel is usually expressed at high levels in sensory nerve fibers where activation mediates the release of pain transmitters, calcitonin gene-related peptide and substance P [56]. While TRP channels are also ionotropic target receptors for endocannabinoids [57], they are also signaling partners with G-protein-coupled receptors [58]. After the identification of TRPA1 as a target receptor for THC, we investigated IL-6, IL-8, and MMP-3 production and proliferation of RASF in response to THC. We found that THC reduced cytokine levels only at the highest concentration and this effect was not antagonized by a TRPA1 inhibitor. This suggests an off-target effect, since the CB_1_ antagonist rimonabant did not reverse the reduction induced by THC. CB_2_ was also not involved, as we previously screened several CB_2_ agonists and antagonists (data not shown) for their ability to modulate cytokine expression in SF and found no effect. This is in line with results from Fechtner et al., who showed that CB_2_ ligands had negligible effects but deletion of the receptor had a proinflammatory impact on SF [32]. In addition, cell viability was impaired at high THC concentrations and we also observed extensive cell death, suggesting a cytotoxic effect by THC. Furthermore, given the affinity of THC at CB_1_ and CB_2_ [59], it is unlikely that only high micromolar concentrations of THC would affect cytokine production. Similar to THC, we already showed a reduction in cytokine production and cell viability by CBD [47]. At high CBD concentrations, this cannabinoid elicited the assembly of the mitochondrial permeability transition pore, which entailed cell death [47], and such a mechanism might also be employed by THC. Concomitantly with the reduction in cytokine levels, we also found reduced proliferation of RASF in response to THC, which demonstrates that cell death at least contributes to reduced cytokine levels.

In RA, lymphocytes and macrophages activate SF in synovial tissue, promoting them from a bystander to an active participant in the inflammatory process [60] and, therefore, we investigated the impact of THC on healthy PBMC alone or in co-culture with RASF. While IL-6 was increased in PBMC co-culture compared to monoculture, IL-10 and TNF levels were lower in co-culture. While IL-6 is mainly produced by RASF, IL-10 and TNF are exclusively produced by PBMC, and RASF have an inhibitory influence on production of these cytokines. A functional repression of B lymphocytes by RASF has also been confirmed in a study by Storch et al. [61]. We found that THC reduced TNF production by PBMC only in co-culture, suggesting a combined inhibitory influence of RASF and THC, while IL-10 was reduced in monoculture. Many studies showed an inhibitory effect of THC on lymphocyte function, but these studies did not investigate cytokine production [62,63,64,65]. In PBMC monoculture, THC had a significant effect on immunoglobulin G production without an activating stimulus (e.g., CpG). This suggests that THC in high concentrations is able to support T or B cell activation directly. In fact, it has been shown that T cells express TRPA1, and activation increased T cell activation and calcium influx [66]. This is in contrast to published findings that CB_2_ mediates the immunosuppressive function of cannabinoids and THC via several different mechanisms [67,68,69], but necessary THC concentrations to activate CB_2_ are much lower than those needed for TRPA1 activation. In line with this, THC in high concentrations has been found to modulate cytokine levels independent of cannabinoid receptors [34]. In general, it is yet unclear whether immunoglobulin production is enhanced or suppressed by cannabinoids. In mice, it has been shown that antigen-specific immunoglobulin M (IgM) and G (IgG) production are not affected by CB_1_ and CB_2_ deficiency [70]. Immunoglobulin E (IgE) production was enhanced by THC in a non-cannabinoid-receptor-dependent fashion [71], and class switching from IgM to IgE was supported by CB_2_ activation in isolated murine B cells [72]. Due to these differential effects of cannabinoids on immune cells, it is difficult to predict whether THC has a stimulatory or inhibitory effect on the immune response in general. In fact, THC impairs the differentiation of monocyte-derived dendritic cells [65], regulates micro RNA relevant for T cell differentiation [73] or inhibits STAT3 phosphorylation [74], and none of these effects might regulate cytokine production in our experimental setting. In addition, THC has several cannabinoid-receptor-independent effects [34,64] and these might be different in every cell type. Single-cell transcriptomic analyses of PBMC revealed that THC does not suppress immune reactions in general but interferes with pro- and anti-inflammatory innate and adaptive pathways of the immune system [75], which might be counterbalanced in our setting.

## 5. Conclusions

Besides CB_1_ and CB_2_, we identified TRPA1 as a target receptor of THC that increases intracellular calcium levels and drug uptake in RASF. Although RASF cytokine production was not altered at relevant concentrations, THC might nevertheless be an important adjunct therapy option in RA, although endocannabinoid degradation inhibitors might be even more suitable, since endocannabinoids only elicit psychotropic effects at high concentrations [76], making the side effect profile more benign compared to THC. THC, on the one hand, might increase effective drug concentrations in target cells when administered with an antirheumatic drug (as demonstrated by PoPo3 uptake). On the other hand, THC might be used to treat comorbidities in RA and, in fact, it has been shown that THC ameliorates depression, sleep disturbances and pain [77,78]. The influence of THC on RA comorbidities might be mediated by the sympathetic nervous system, since CB_1_ receptors control sympathetic outflow centrally and peripherally [9,79]. TRP channels in the nervous system increase excitability and neurotransmission [80], and, therefore, it might be possible that these channels also mediate the release of soluble mediators by peripheral cells, such as fibroblasts or tyrosine hydroxylase positive cells [81] that are increased in chronic inflammation. Although speculative, THC might trigger the release of catecholamines from tyrosine-hydroxylase-positive cells or RASF via TRPA1 activation, which would increase the sympathetic tone and provide anti-inflammatory effects via β adrenoceptor signaling [82,83]. While TRPA1 activation is the main mechanism by which RASF are modulated by THC, PBMC also respond to CB_1_ or CB_2_ stimulation [84]. Since THC reduced TNF at a physiological concentration of 1 µM [85], it might be a cannabinoid-receptor-driven effect. In addition, endocannabinoid levels might be increased by THC [86] in co-culture, since RASF and lymphocytes contribute to their production [29,49] and they might also be responsible for the observed inhibitory effects. In addition, THC’s complex effects on the immune response (e.g., differentiation, intracellular signaling and activation of immune cells) might also support its use in chronic inflammatory conditions, such as RA.

## Figures and Tables

**Figure 1 biomedicines-10-01118-f001:**
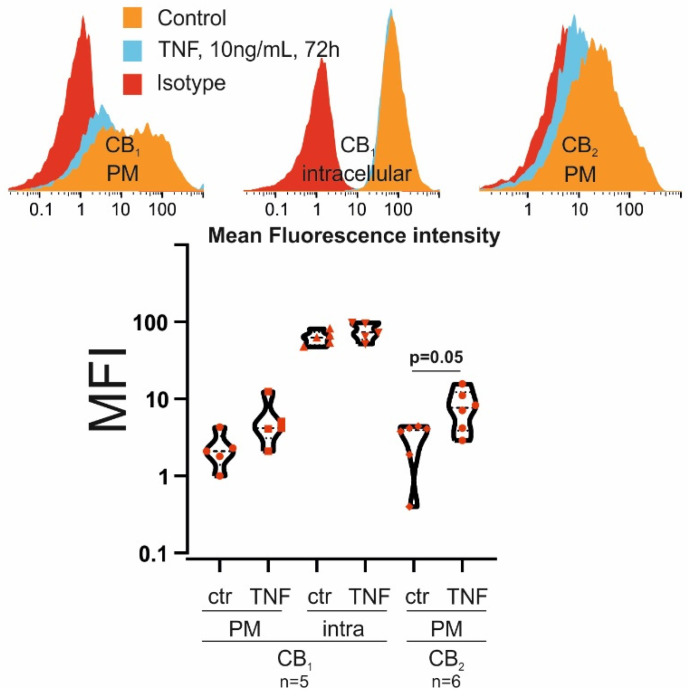
**Flow cytometric detection of CB_1_ and CB_2_ in and on RASF.** RASF were incubated with TNF (10 ng/mL) for 72 h, and CB_1_ and CB_2_ levels were determined thereafter. Upper panel: histogram; detection of CB_1_ at the plasma membrane (PM) and intracellularly and CB_2_ at the plasma membrane. Lower panel: violin plots; quantification of CB_1_ and CB_2_. *t*–test was used for comparisons. *p* = 0.05 was the level of significance.

**Figure 2 biomedicines-10-01118-f002:**
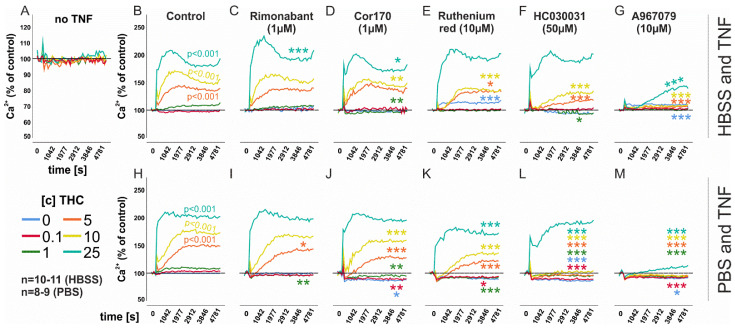
**Mean intracellular calcium level changes in RASF in response to THC.** (**A**) Intracellular calcium mobilization without TNF pre–stimulation. (**B**,**C**) Intracellular calcium level regulated by THC (0.1−25 µM) with TNF (10 ng/mL) pre–stimulation for 72 h and extracellular calcium (HBSS; (**B**–**G**)) or without calcium (PBS; (**H**–**M**)). *** *p* < 0.001, ** *p* < 0.01, * *p* < 0.05 for differences between antagonist treatment and control (THC only). (**B**,**H**) Comparisons of different THC concentrations versus control (no THC). Significant values are given in the graph. ANOVA with Bonferroni post hoc test was used for all comparisons. Rimonanbant, CB_1_ inverse agonist; COR170, CB_2_ inverse agonist; A967079, TRPA1 antagonist; HC-030031, TRPA1 antagonist; RR = Ruthenium Red, general TRP inhibitor.

**Figure 3 biomedicines-10-01118-f003:**
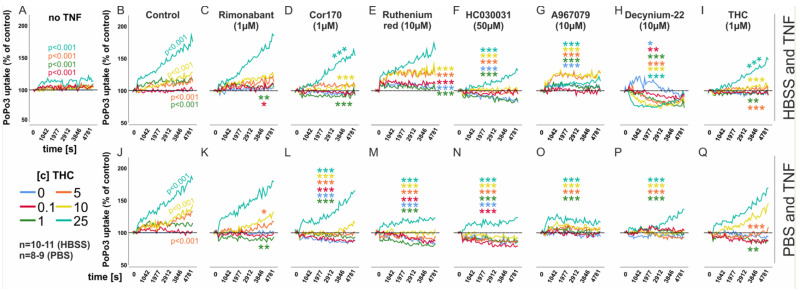
**Mean PoPo3 uptake by RASF in response to THC.** (**A**) PoPo3 uptake without TNF pre–stimulation, (**B**,**C**) by THC with TNF (10 ng/mL) pre–stimulation for 72 h and extracellular calcium (HBSS; (**B**–**I**)) or without calcium (PBS; (**J**–**Q**)). *** *p* < 0.001, ** *p* < 0.01, * *p* < 0.05 for differences between antagonist treatment and control (THC only, Figure 2B,J). (**B**,**J**) Comparisons of different THC concentrations versus control (no THC). Significant values are given in the graph. ANOVA with Bonferroni post hoc test was used for all comparisons. Rimonanbant, CB_1_ inverse agonist; COR170, CB_2_ inverse agonist; A967079, TRPA1 antagonist; HC-030031, TRPA1 antagonist; RR = Ruthenium Red, general TRP inhibitor, Decynium-22, organic cation transport inhibitor.

**Figure 4 biomedicines-10-01118-f004:**
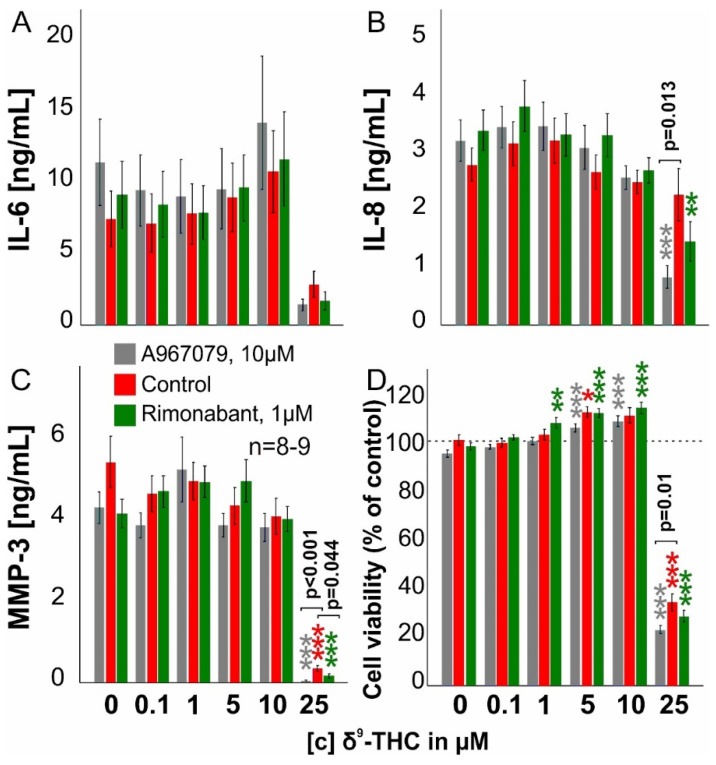
**IL-6, IL-8, and MMP-3 production and live cell number after 72 h incubation with TNF and an additional 24 h of THC.** RASF were incubated for 72 h with TNF (10 ng/mL). After wash-off, RASF were challenged with antagonists for 30 min, followed by THC (0.1–25 µM) addition for 24 h. ANOVA was used for all comparisons vs. control w/o THC. (**A**–**C**) IL-6, IL-8, and MMP-3 production after 24 h challenge with THC. (**D**) RASF cell number after 24 h challenge with THC. Significant differences between THC in different concentrations are depicted as * *p* < 0.05, ** *p* < 0.01, *** *p* < 0.001. ANOVA with Bonferroni post hoc test was used for all comparisons. Differences between THC with or without rimonabant or A9607079 are shown in the graph.

**Figure 5 biomedicines-10-01118-f005:**
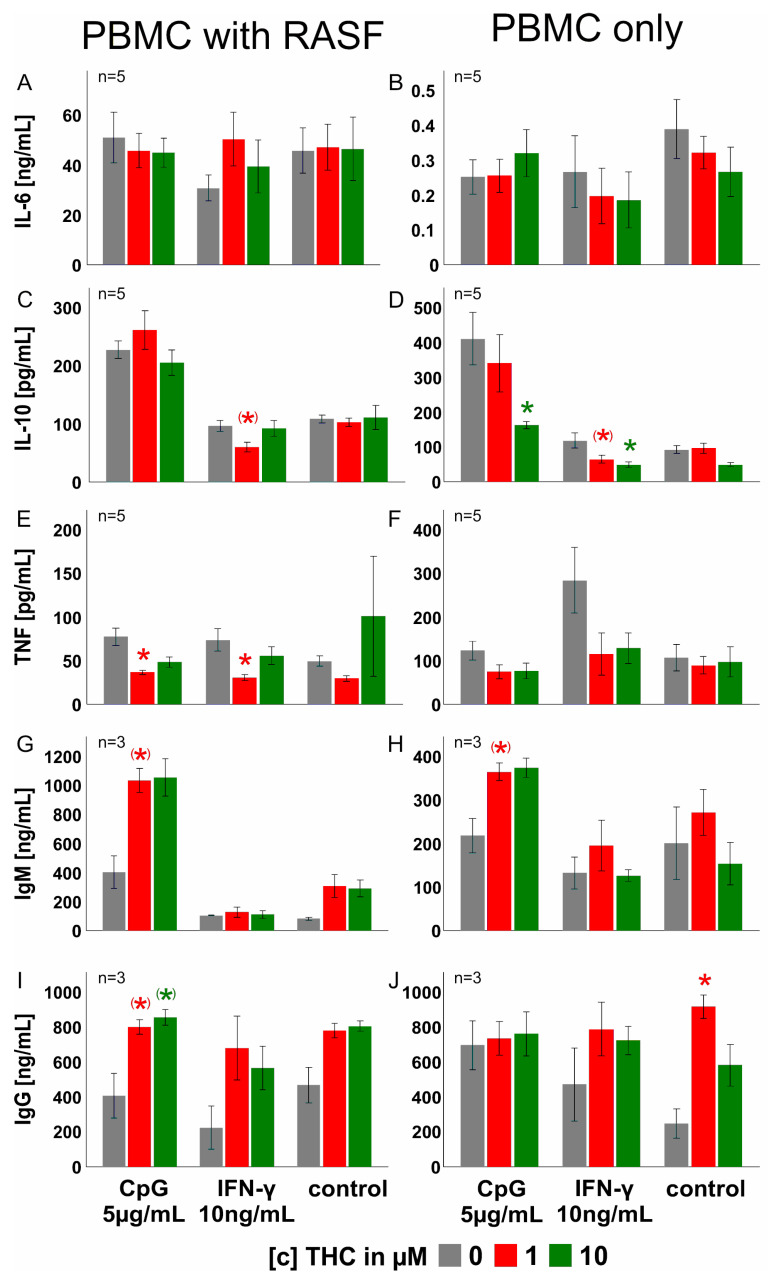
**Cytokine production by human PBMC monoculture and co-culture with RASF in response to THC.** (**A**,**B**) IL-6; (**C**,**D**) IL-10; (**E**,**F**) TNF; (**G**,**H**) immunoglobulin M (IgM) and (**I**,**J**) immunoglobulin G (IgG) production by PBMC and PBMC/RASF co-culture over 7 days. Cells were concomitantly stimulated with THC and the respective activation stimulus (IFN-γ or CpG). In co-culture experiments, RASF were stimulated with 10 ng/mL IFN-γ 72 h prior to PBMC addition to induce MHC II expression and induce an allogeneic T cell response. ANOVA with Bonferroni post hoc test was used for all comparisons. * *p* < 0.05.

## Data Availability

The datasets used and/or analyzed during the current study are available from the corresponding author on reasonable request.

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
