# Peer review of "Impact of Δ9-Tetrahydrocannabinol on Rheumatoid Arthritis Synovial Fibroblasts Alone and in Co-Culture with Peripheral Blood Mononuclear Cells"

_biomedicines, 2022, doi:10.3390/biomedicines10051118_

Round 1

Reviewer 1 Report

The authors explored the effects of THC in synovial fibroblasts from patients with RA and PBMCs from healthy donors. They concluded that the presumable use of THC for treatment of inflammation should include drug titrating due to pro- and anti-inflammatory effects of THC.

Comments

  1. All the abbreviations should be disclosed at first use including Abstract.
  2. Abstract: Every concentration of THC that was applied should be indicated throughout the text and in the figure legends.
  3. Lines 79-80: This sentence is not clear. This should be clarified.
  4. Lines 86-87: The importance and functions of all the compounds should be disclosed.
  5. Lines 138-140: The authors should define whether they analyzed viability or proliferation using the indicated test system. Promega corporation designates this test only as a CellTiter-Blue Cell Viability Assay. 
  6. Fig. 4: As cell viability (which the authors designate as proliferation) was significantly lower at THC concentration 25uM, the results (and discussion) related to this THC concentration should be removed from the manuscript.

Reviewer 2 Report

The authors propose an interesting study testing THC on synovial fibroblasts of patients with rheumatoid arthritis (RASF) and peripheral blood mononuclear cells (PBMC). The results provide important insights into the cannabinoids potential for the treatment of RA. This study represent a initial investigation on the use of THC as adjuvant therapy for RA.    Line 79, please specify the therapies of all patients because they are reported only for 12 patients.   Line 142-147, please cite the references corresponding to the statistical tests used.
